# The Effects of Foods Embedded in Entertainment Media on Children’s Food Choices and Food Intake: A Systematic Review and Meta-Analyses

**DOI:** 10.3390/nu12040964

**Published:** 2020-03-31

**Authors:** Victoria Villegas-Navas, Maria-Jose Montero-Simo, Rafael A. Araque-Padilla

**Affiliations:** Department of Management, Universidad Loyola Andalucía, Escritor Castilla Aguayo St., 4, 14004 Cordoba, Spain; jmontero@uloyola.es (M.-J.M.-S.); raraque@uloyola.es (R.A.A.-P.)

**Keywords:** foods embedded, entertainment media, meta-analysis, children, choice, intake

## Abstract

While watching or playing with media, children are often confronted with food appearances. These food portrayals might be a potential factor that affects a child’s dietary behaviors. We aimed to comprehensively expound the effects of these types of food appearances on dietary outcomes of children. Our objectives were to synthetize the evidence of the experiments that study the effects of foods embedded in children’s entertainment media throughout a systematic review, to conduct two meta-analyses (food choice and intake) in order to quantify the effects, and to examine to what extent the effects of foods embedded in entertainment media varies across different moderating variables. We conducted a systematic search of five databases for studies published up to July 2018 regarding terms related to children and foods embedded in entertainment media. We identified 26 eligible articles, of which 13 (20 effect sizes) and 7 (13 effect sizes) were considered for a meta-analysis on food choice and intake, respectively. Most of the studies were assessed as having a middle risk of bias. Overall, food being embedded in entertainment media is a strategy that affects the eating behaviors of children. As most of the embedded foods in the included studies had low nutritional values, urgent measures are needed to address the problem of childhood obesity.

## 1. Introduction

“Ensuring healthy lives and promote well-being for all at all ages” is the third Sustainable Development Goal (SDG) of the 2030 agenda, which was established by the United Nations. Healthy lives include the acquisition of healthy habits. The World Health Organization (WHO) [1] points out several steps to develop a healthy lifestyle: the first step emphasizes the need to eat a nutritious and varied diet. Eating healthy foods helps to maintain a balanced weight and helps to prevent obesity. However, obesity represents a global issue [2] and its prevention starts in childhood. The prevalence of childhood obesity is rising worldwide [3]. Longitudinal studies show that childhood obesity is associated with non-communicable diseases (NCDs), such as cardiovascular, metabolic or pulmonary diseases [4], and with a lower life expectancy [5].

The eating habits of children are not easily influenced; there are multiple factors involved [6]. Regarding the environmental factors, children are surrounded by different food messages that affect their dietary patterns. These food messages come from different sources [6], such as parents [7], schools [8], peers [9] or media [10]. The media environment plays a key role in shaping eating habits, as children spend much of their time in front of a screen—including TV, Desktops, Laptops, Mobile phones, or Handheld Game Consoles [11,12]. The technological revolution has been favoring the exposure of children to food portrayals that do not interrupt the storyline of the corresponding entertainment media or that cannot be skipped by audiences. This current environment leads us to study what we have labeled “foods embedded in entertainment media”.

When foods are embedded in entertainment media, they appear in a natural and spontaneous atmosphere, in comparison to, for example, commercials. Due to its authenticity, children are less aware of the potential persuasiveness of these food portrayals, as compared to ads [13]. Additionally, enjoyment is usually present as children choose and search for the entertainment media they like. Children are exposed to foods that are embedded in entertainment media from a variety of circumstances: while watching TV (in cartoons, sitcoms or TV shows), going to cinema (in movies) or using different digital platforms, playing online or non-online games, visiting brands websites and playing advergames, among others.

Previous studies, based on content analyses, have found that the prevalence and prominence of foods embedded in entertainment media, such as movies [14], advergames [15] or cartoons [16] are high, and a substantial portion of these foods have low nutritional values. Content analyses are highly relevant, when combined with studies that explore the potential effects of foods embedded in entertainment media, to the dietary outcomes of children.

Published systematic reviews and meta-analyses have provided insight about those potential effects. As far as we know, those reviews have focused on whether some specific entertainment formats, such as videogames [17] and advergames [18,19], and whether marketing products in general (food embedded with a clear persuasive intention) [20,21], are impactful tools to modify the eating behaviors of children. However, we have not found systematic reviews or meta-analyses that include a more ample vision of foods embedded in entertainment media. The present study will lead us to have a more global perspective on these food portrayals, not only by measuring their effects, but also by studying different moderating variables that might influence the impact of the embedded foods on the dietary behaviour of children.

There are certain aspects that are recurrent in the literature, as these constitute key elements when measuring the effects of foods embedded in entertainment media. First, as the present study is concerned with a specific type of product, that is, foods, it is essential to differentiate between the types of food according to their health benefits. Sadeghirad and colleagues [21] made that distinction, in their meta-analysis of food marketing, and found that the marketing strategies of low nutritional value foods were associated with higher caloric intake, in contrast to high nutritional value foods (which were not significantly associated with intake). Exploring whether low or high nutritional value foods have similar or different effects is highly relevant to establish policy measures that are aimed at addressing the problem of childhood obesity.

Second, foods embedded in entertainment media might appear, whether associated with a determined brand (branded) or not (non-branded). In branded foods studies, children are usually confronted with foods that belong to the same food category (sweets, chocolates, snacks or soft drinks). For example, in Naderer, Matthes and Zeller [22], children had to choose between three different chocolates: M&M’s (target brand), Maltesers, or Dragee Keksi. However, in studies that analyzed non-branded foods, children were confronted with different types of foods. For instance, in Folkvord, Anastasiadou and Anschütz [23] found that children that were presented pieces of mandarins, apples, bananas, and grapes. While the type of food, according to the appearance of brands (branded or non-branded), has not been studied previously in the literature, and we consider that this distinction is relevant as there is a qualitative leap between both types of foods (branded and non-branded) when exploring the effects of foods embedded in media.

Third, other characteristics are intrinsically associated with how embedded foods appear; these are the integration characteristics [14] (also known as execution factors [24] or composition characteristics [25]). The integration characteristics that the present meta-analysis will consider are two of the most studied integration characteristics: modality and plot connection [26,27,28]. Regarding modality, previous studies suggest that foods that are embedded bimodally (that is, portrayed both visually and verbally) are more likely to have effects in terms of recall, recognition and behavioral outcomes, such as increasing choice [26], than unimodal foods (that is, those that are portrayed whether visually or verbally). However, plot connection refers to the degree in which the food appearance is connected to the storyline. A food portrayal is connected to the plot when the food appearance is essential, that is, in the case where the food was removed, the storyline would be senseless [16]. Similar to bimodal embedded foods, it is assumed that foods connected with the plot are associated with greater food effects, as children pay more attention to these portrayals, which integrate them in a deeper way [22].

Fourth, watching (for example, movies, cartoons, and sitcoms) and playing (for example, videogames or advergames) entertainment media constitute different activities: the second requires active interaction, while the first does not (passive interaction); previous meta-analyses have explored this distinction [21], and the present review will take this aspect into account.

Fifth, the age of children is another variable that has been extensively studied in the literature [13,22,29,30]. Whereas some studies did not find moderating effects of children’s age [13,22], others did [29,30]. In general, it is assumed that the cognitive abilities of children, as well as persuasion knowledge, improve as children age [31,32].

Therefore, given the influence that foods embedded in entertainment media have on the eating behaviors of children, we aimed to systematically review the studies that measure the effects of the food portrayals that target children. Our objectives were to: (1) synthetize the evidence of the experiments that study the effects of foods embedded in entertainment media; (2) conduct meta-analyses to determine the effects of these portrayals on food choice and food intake; and (3) examine to what extent the effects of the aforementioned foods embedded in entertainment media vary across the moderating variables.

## 2. Materials and Methods

The current systematic review was reported in accordance with the Preferred Reported Items for Systematic Review and Meta-analysis (PRISMA) statement [33] (Appendix A). The protocol of this review was retrospectively registered with PROSPERO in July 2019 (ID CRD42019125907) [34].

### 2.1. Search Strategy

The following five electronic databases were searched during July 2018: Academic Search Ultimate, Business Source Ultimate, MEDLINE, PsycINFO and PubMed. The language of the search was limited to English. No date restriction was placed on the search. The search strategy is available in Appendix A (keywords had a unique domain: foods). Additionally, we undertook forward and backward citation tracking from the identified papers in order to collect articles that use different terminology. The database search results were imported into Mendeley Desktop software and duplicates were removed.

### 2.2. Eligibility Criteria

Studies were eligible for inclusion in the systematic review if (i) participants were children aged under 12 years (as children aged above 12 years are considered adolescents [35] and they were not the focus of the present study); (ii) interventions included foods exposure in entertainment media targeted at children (the media considered in the inclusion criteria were both watching and playing entertainment media, excluding social media); (iii) they included parallel comparison groups that were not exposed to embedded foods or that were exposed to different embedded foods to the intervention condition; and (iv) reported outcomes included either food choice or food intake.

Eligibility criteria for the meta-analyses were more restrictive—not only did the studies need to fulfil the criteria for the systematic review, they also had to include a suitable comparison group (participants who were not exposed to the object of the embedded foods study).

### 2.3. Study Selection

Double independent searching for eligible studies, by viewing titles and abstracts, was conducted by two researchers (IC and VV). The full texts of all potentially relevant studies were obtained and assessed against the inclusion criteria by IC and VV. Disagreements about the eligibility of a study was resolved by discussion and consensus by a third author (MM).

### 2.4. Data Extraction and Quality Assessment

Relevant information was extracted: general study information (authors’ names, publication year, journal, and study location), study population details (sample size, age, and ratio of female vs. male), name of the embedded target food(s), entertainment media used to embed the food, intervention components, behavioral outcomes (differentiating between choice and intake) and information to assess the risk of bias.

Additionally, data was codified independently by IC and VV for the type of food, according to its healthiness, type of food according to the appearance of brands, plot connection, modality, type of entertainment media and quality assessment. When codifying the type of food according to its healthiness, three categories were considered, following the World Health Organization (WHO) [36] criteria: low nutritional value foods (those that are high in fats, sugars or salts such as snacks, sweets or soft drinks), mixed nutritional value foods (such as dairy products, white meat, fish, eggs, and legumes) and high nutritional value foods (such as fruits and vegetables). When codifying the type of food according to the appearance of brands, two categories were considered (branded and non-branded foods). If one of the variables could not be encoded (because there was not enough information in the article or because we could not contact the respective authors of the included studies), they were not categorized. To codify the plot connection, the contribution that the embedded food made to the storyline was differentiated between essential (plot connected) or not essential (plot disconnected). Modality was codified by differentiating between unimodal (the food is only seen or only mentioned) or bimodal (the food is both seen and mentioned). The type of entertainment media codification showed the difference between those media that require only watching and those media that combine watching and playing. Age was codified according to the major cutoff points in terms of child knowledge of advertising [31,37]: up to 5 years, from 6 to 7 years and from 8 to 12 years. When further information about a study was required, the corresponding authors were contacted.

We used the Cochrane risk of bias tool [38] to assess the quality of the studies. The quality assessment was codified according to the risk of bias of each: low (risk of bias punctuation up to 3), middle (risk of bias punctuation from 4 to 5) and high (risk of bias punctuation higher than 5).

### 2.5. Data Synthesis and Statistical Analysis

To assess the effects of foods embedded in entertainment media on the food behavioral outcomes of children, we used two effect measures: risk ratio (RR) and mean difference (MD). The food choice of children was reported as the percentage of children who preferred the object of the target food study. We treated this as a dichotomous variable (yes/no) and pooled eligible trials using the RR and the corresponding 95% confidence intervals (CIs). We calculated the MD, and its corresponding 95% CIs, for dietary intake, which were reported as kilocalories (kcal) of foods/beverages consumed during or immediately after the experiment. As the MD can only be used when all the outcomes are measured on the same scale, we transformed the data from the results that were not reported in kcal [39] (e.g., grams). Data analysis was conducted using the random effects models with RevMan and Comprehensive Meta-Analysis software. An a priori assumption guided this choice. As the studies were not identical, we assumed that a true effect size was not possible. Statistical heterogeneity was evaluated using the *I*^2^ statistic. In addition, we calculated the Q statistic and its *p* value.

Moderator analyses were conducted based on the codified data. We did not conduct the moderator analysis on the type of food according to its healthiness on food intake, because kcal might not be a sensible outcome measure, given that low and high nutritional value foods tend to be different in kcal [36]. It is relevant to highlight that a different trial within the study was considered when the experimental group was composed of children from different regions, or when the experimental exposure varied between conditions, either for the embedded food or for the execution variables used. Additionally, we considered that, to compare the categories within a moderator analysis, each category had to be composed of at least two trials. We tested for interaction using a chi-square significance test [40] for moderator analyses. Publication bias was examined throughout funnel plots in order to detect possible reporting biases in the meta-analysis via visual inspection, and by using both Egger’s regression method [41] and a trim-and-fill analysis [42].

## 3. Results

In total, 1624 unique studies were identified through database searches, of which, 26 [13,22,23,26,29,30,39,43,44,45,46,47,48,49,50,51,52,53,54,55,56,57,58,59,60,61] were deemed eligible for inclusion. Compared to the studies included in similar, previous systematic reviews [62] and meta-analyses [17,18,19,20,21], the present meta-analysis contained 12 extra studies [22,23,26,29,30,49,51,52,55,56,58,60]. The PRISMA flow diagram is provided in Figure 1.

The inter-coder agreement reached during the eligibility phase was substantial (kappa = 0.79, 95% CI 0.59–0.97). The observed agreement was 90.2% (37/41 decisions). While the reasons for exclusion were not mutually exclusive, when deciding whether the article had to be included or not, we considered three reasons, which are shown in order in Figure 2. The inter-coder agreements reached during codification varied, ranging from the lowest kappa obtained (kappa for modality = 0.52, 95% CI 0.13–0.92) to the highest kappa obtained (kappa for branding = 1, 95% CI 1–1). A summary of the studies is provided in Appendix A.

### 3.1. Results of the Systematic Review

Table 1 presents a synthesis of the characteristics of the included studies in the systematic review and also separately shows a summary of the characteristics of the included comparisons for both the meta-analyses on food choice and food intake. A quality assessment summary is represented in Figure 2 (Appendix A).

Overall, and according to Appendix A, foods embedded in entertainment media reported significant effects on food choice [13,22,26,30,43,50,51,52,55,59,60,61]. However, six studies did not find significant effects of foods embedded in entertainment media on food choice [49,53,54,56,57,58]. Two of these studies inserted foods digitally (that is, subtly) [49,56], and there was another study where the mean age of the participants was approximately 12 years (that is, the older children group) [58]. Additionally, one study reported significant effects when there was an interaction between the type of food and the childrens age: low nutritional value foods embedded in cartoons were more likely to present effects when children were younger than 9 [29].

Most of the studies about foods embedded in entertainment media in the systematic review were also likely to present effects on food intake [23,44,45,46,48]. However, there were two studies that did not report significant effects on food intake [47,60]. While Brown and colleagues [60] found significant food choice effects of foods embedded in the analyzed movie, they did not find effects of the same embedded foods on intake. In addition, in Folkvord and colleagues’ study [47], the foods embedded (advergames) did not influence the later intake. However, the intake measures were not taken immediately after advergame exposure [47]. However, in Harris and colleagues’ study [39], significant effects were found when comparing low and high nutritional value advergames, but not when comparing each type of advergame condition with the control group.

### 3.2. Results of the Meta-Analyses

Of the 26 articles included in the systematic review, 13 and 7 articles were considered for meta-analysis on food choice and food intake, respectively (which resulted in 20 and 13 comparisons, respectively).

#### 3.2.1. Food Choice

Considering the 21 comparisons of food choice, children exposed to foods embedded in entertainment media had a greater risk of choosing the foods embedded, as compared to children who were not exposed to such foods (RR = 1.41, 95% CI 1.14 to 1.75, *p* < 0.001, *I*^2^ = 74%; Figure 3; Table 2).

Table 2 reports the moderator analyses of foods embedded in entertainment media on choice. Considering the *p*-value interaction of the moderator analyses on choice, only the modality shows a statistically significant moderator effect (*p* = 0.039). The effects of foods embedded in entertainment media are greater for bimodal, rather than for unimodal portrayals. However, there is a substantial, unexplained heterogeneity between the trials within the bimodal moderator category (*I*^2^ = 79.74%). Therefore, the validity of the effect estimate for modality is uncertain, as individual trial results are inconsistent.

However, Table 2 shows the moderator analyses with non-significant *p*-values for interaction, but significant *p*-values for one of the included categories within the moderator analyses. Those are the cases of the moderator analyses of the type of foods (significant *p*-value for low nutritional value foods), branding (significant *p*-value for branded foods), plot connection (significant *p*-value for foods connected with the plot), entertainment media activity (significant *p*-value for the watching activity) and quality assessment (significant *p*-value for middle risk of bias studies). In all of these cases, there were a predominant number of trials within the specific category, as compared to the contrasting one (for example: 18 trials of low nutritional value foods against 2 trials of mixed nutritional value foods). Additionally, according to the *I*^2^ value, those categories showed important, unexplained heterogeneity.

Lastly, when analyzing age, the *p*-value for interaction was also non-significant. However, children aged from 6 to 7 years and from 8 to 12 years showed a significantly increased risk of choosing the embedded foods.

#### 3.2.2. Food Intake

According to the 13 comparisons included in the meta-analysis of food intake, children exposed to foods embedded in entertainment media significantly increased the kcal consumed during or shortly after the experiment, as compared to children who were not exposed to the embedded foods (MD = 25.51, 95% CI 14.37 to 36.66, *p* < 0.000001, *I*^2^ = 88%; Figure 4).

We could only assess the planned moderator analyses for age and quality assessment, as we had insufficient data to assess the rest of the moderators (we determined that one trial was not enough to compare within the moderator category), as shown in Table 3.

The test for age moderator differences suggests that there is a statistically significant moderating effect (*p* = 0.016). While both age groups reported significant effects in terms of MD, the effects of foods embedded in entertainment media on intake are greater for children aged from 6 to 7 years than for children over 8 years. However, there is a substantial, unexplained heterogeneity (*I*^2^) between the trials of children aged from 8 to 12 years.

Lastly, the test for the moderator quality assessment indicates that there is no statistically significant moderator effect (*p* = 0.259), suggesting that the study risk of bias does not modify the effects of foods embedded in entertainment media. However, significant *p*-values were found for the categories considered in this moderator analysis: low and middle risk of bias (the low category being the one which reported the highest MD).

### 3.3. Publication Bias

Funnel plots are a primary visual tool to explore publication bias in meta-analyses and they show the distribution of effect sizes [63]. These graphs are named funnel plots because of the symmetric shape they are expected to have. At the bottom of the funnel plot, we can find those studies that are smaller (larger standard error), and while we are approximating to the top, we find those studies that have a larger sample size. Additionally, it is expected that each study (represented by the effect size) on one side of the funnel plot has “its equivalent” on the other side. However, when funnel plots are not symmetric, it might be possible that publication bias exists. Publication bias is frequent when it is not possible to find published studies with small sample sizes that simultaneously found no or only small measure effects [64]. Taking these considerations into account, a visual inspection of funnel plots was performed for both food choice and food intake (Figure 5 and Figure 6).

Both figures represent funnel plots with imputed studies (in this study, with imputed comparisons) after adjusting for publication bias. The included studies are illustrated as white dots, while the imputed studies are the dots colored in black. Imputed studies try to provide symmetry to the graph [42]. While the funnel plot, based on food choice, did not present imputed studies (Figure 5), the funnel plot based on food intake did (Figure 6). Specifically, six studies were imputed, which suggests that there is evidence of asymmetry. In addition, the white diamonds represent the overall effect of the included studies, while the black diamond represents the effect once imputed studies have been considered. There was no variation in the diamonds (black and white) of the funnel plot, based on food choice (Figure 5), but there was a variation in the diamonds of the funnel plot, based on food intake (Figure 6): the new diamond (black) includes the null value (which would mean a weakening or even disappearance of the overall effect).

Lastly, we formally examined publication bias with the use of Egger’s test [41]. The Egger’s test *p* value was 0.252 for food choice and 0.02 for food intake, which suggested that there is evidence of asymmetry in the case of studies that measure food intake but could be attributed to publication bias.

## 4. Discussion

In the present systematic review, 26 articles from studies that experimentally manipulated the exposure of foods embedded in children’s entertainment media and measured behavioral outcomes were identified. We conducted two meta-analyses, according to the behavioral outcomes measured: food choice (13 articles, which contributed 21 effect sizes) and food intake (7 articles, which contributed 13 effect sizes). Overall, most of the studies focused on embedded foods that had low nutritional values, and were branded, connected with the plot and bimodal.

The results of the present systematic review on food choice suggest that foods that are embedded subtly, such as those inserted digitally [49,56], are less likely to present effects than those embedded more prominently, such as those that are connected with the plot [65] or that show interactions with the character(s) [22]. The results also show that, as children grow up, they might be less persuaded by foods embedded in entertainment media [58]. In addition, it might be possible that it is the interaction of several variables that makes foods embedded in entertainment media more impactful: the combination of low nutritional value foods targeted to children younger than 9 years old [29].

The direction of the results of foods embedded in entertainment media on food choice or intake might differ, as Brown and colleagues [60] found effects of the embedded foods on food choice, but not on food intake. Additionally, most of the outcomes measured in the included studies were taken immediately after exposure of the embedded foods. However, when measuring with a more ample timeframe, the effects were not found [47]. This is highly relevant as children might refrain from their impulses to eat the embedded foods if they do not have direct access to those foods.

Regarding the meta-analyses, we found that the exposure of children to foods embedded in entertainment media increased the likelihood of choosing the embedded foods (food choice). Moreover, children were more likely to increase their consumption of the foods (food intake), either during or shortly after exposure to the embedded foods. Therefore, embedding foods in children’s entertainment media seems to be a strategy that influences the food behaviors of children.

As indicated beforehand, previous literature (individual studies) has explored what makes foods embedded in children’s entertainment media more impactful in regards to affecting children’s food behavior. In the present study, moderator analyses were performed to explore those characteristics that might affect the effects of foods embedded in children’s media.

In the present meta-analysis, when measuring food choice, the moderator type of food did not reveal significant differences in terms of interaction. However, when analyzed separately, low nutritional value foods were associated with an increase in the risk of choosing the embedded foods. These results are in consonance with previous meta-analyses that focused on low nutritional value foods exposure targeted to children in regards to behavioral outcomes [18,20,21]. For example, in Folkvord and Van’t Riet’s [18] study, it was found that advergames that promote low nutritional value foods induced unhealthy eating behavior among children. In addition, Boyland and colleagues [20] found that exposure to unhealthy food advertising was associated with higher food intake. Sadeghirad and colleagues’ meta-analysis [21] showed that unhealthy food marketing leads to an increased risk of children selecting the commercial’s products. Overall, our study provides another contribution to the effects of low nutritional value food portrayals targeted at children. There are several reasons why low nutritional value foods are impactful: they are enjoyed while eating [66], they elicit implicit affective evaluation [67], they elicit high attention [68], among others.

However, branding did not moderate the effects of foods embedded in entertainment media. However, when considered separately, branded embedded foods were significantly associated with greater risks of choosing the foods, as compared to non-branded foods. These results might be due to the higher engagement that branded foods included in the studies had in the meta-analysis on food choice, as compared to non-branded foods. The marketing food industry spends large amounts of money in communicating their products and this is translated into appealing and child-oriented food messages [69].

As far as we know, plot connection has not been explored in previous meta-analyses. While significant differences were not found in terms of interaction, embedded foods connected to the plot were associated with increased risks of selecting the foods, as compared to those that were disconnected. Embedded foods connected to the plot might work as prominent stimuli that facilitate recognition and lead to increased positive evaluations [28].

Regarding modality, our results showed significant differences in terms of interaction: bimodal embedded foods were associated with greater risks of choosing the foods than the unimodal ones. While studies on adults showed that bimodal embedded foods might be self-defeating when changing food preferences [27], it seems that, in the case of children, bimodal portrayals constitute an impactful tool. According to Dual Coding Theory (DCT) [70], modality might work as a hierarchical sequential structure where bimodal embedded foods are more extended than the unimodal ones, making bimodal embedded foods more salient.

We found a previous meta-analysis that explored the distinction between advergames and TV advertising [21] that found non-significant differences in terms of effects. In the present meta-analysis, we explored the moderator type of entertainment media (confronted watching vs. playing activity) and we non-significant results in terms of interaction. However, when analyzed separately, watching activity showed significant effects (as compared to playing activity). These results are in consonance with previous literature that suggests that the activity of playing (active reception and high interactivity) might present equal effects with the activity of watching (passive reception and low interactivity) [71].

Finally, the age of the participants was explored in previous meta-analyses on food marketing [20,21]. A study by Boyland and colleagues [20] examined the differences between children and adults, and found that, in the short term, food advertisements were associated with increases in food intake in the case of children, but not in adults. Similarly, in a study by Sadeghirad and colleagues [21], children under 8 years of age were influenced by food marketing strategies at a conative level (preferences). Our findings suggest that a child’s age might act as a moderating variable; younger children (aged 6–7) were the ones associated with increases in both the measured behavioral outcomes during and after exposure to foods embedded, as compared to older children (aged 8–12). These results are in line with previous studies concerning the influence of age on the persuasive skills developed during childhood, i.e., the younger the child, the less persuasive skills they have [37], and therefore, their strategies to defend against the effects of foods embedded in entertainment media are less developed.

Regarding quality assessment, none of the behavioral measures were significant in terms of their moderator interaction. A previous meta-analysis [21] by Sadeghirad and colleagues found similarly nonsignificant results for food choice, although significant differences were found for dietary intake (low risk of bias studies were associated with greater increases in caloric intake than high risk of bias studies). However, we found significant effects in studies rated with a middle risk of bias on food choice (in contrast to the non-significant effects of studies rated with a high and low risk of bias). Unexpectedly, studies rated with middle quality assessment were more likely to have an effect than those rated with high quality assessment (low risk of bias) on food choice. However, when it comes to food intake, studies rated with low risk of bias reported higher caloric intake than those rated as middle risk of bias, which is consonance with previous meta-analyses [21]. Lastly, it is important to highlight that the outcomes of food choice are more robust than those related to food intake: we did not detect publication bias for food choice but did for food intake. A possible explanation might be that, according to Cruwys, Bevelander and Hermans’ review [9], modeling of food choice is less prominent than modeling of food intake. When it comes to modeling food choice (in contrast with modeling food intake), the pre-existing preferences of people might act as a mediator for the effects of foods embedded in entertainment media. That is, the effects of foods embedded in entertainment media on food choice might decrease or even disappear when the pre-existing preferences of children emerge. That might be the reason why we found more published studies with small sample sizes with no or small measure effects on food choice, as compared to food intake.

As the protocol was not prospectively registered, this constitutes a limitation. Additionally, we did not use a generalist research platform (such as Google Scholar), which might have provided more studies not included in the present systematic review, and we did not include unpublished material. As important, unexplained heterogeneity was obtained for both food choice and food intake, moderator analyses were conducted to explore it. Overall, although moderator analyses did not reveal a unique key characteristic that explained the heterogeneity found, the results of the meta-analyses show a draft about what aspects (characteristics) might be more relevant when measuring the effects of foods embedded in entertainment media. Additionally, it might be possible that some of the moderator analyses were not able to detect differences: a small number of trials and participants in the case of foods that were high in nutritional value, which were non-branded, not connected with the plot and that were associated with the playing activity might be the reason for the undetected differences. Additionally, as the moderator analysis age was calculated according to the mean age of the participants, the results might be affected by the age the standard deviation of some of the included studies. In addition, the meta-analysis included experiments conducted at a specific moment in time, and as Sadeghirad and colleagues [21] noted in their meta-analysis about food marketing, we could not assess the cumulative effects that exposure to the embedded food had on children over their lifetime.

Most of the studies included in the present systematic review have been conducted in recent years and might reflect the relevance that the study of foods embedded in entertainment media is acquiring currently and the increasing concern about how this strategy affects the food behaviors of children and eventually childhood obesity.

The problem of childhood obesity has acquired the label of an ‘epidemic’ in the last decades [72,73]. As previous research indicates, it is widely accepted that food marketing acts as a contributing factor to the problem [18,20,21]. Foods embedded in entertainment media might constitute another marketing strategy (in the case of branded foods) or it might not (non-branded foods). In any case, food portrayals represent a powerful communication tool that affects the eating habits of children. In 2010, the WHO established a set of recommendations on the marketing of foods and non-alcoholic beverages to children [2], and foods embedded in entertainment media was considered to be one of the marketing techniques involved in the problem of childhood obesity. As low recommended foods are prevalent in entertainment media [14,15,16], combined with the results of this present meta-analysis, urgent action is needed to address the problem of childhood obesity. We recommend that policymakers pay special attention to the target age of the entertainment media (as younger children are more vulnerable to the influence of foods embedded in entertainment media) when designing public policies. We also recommend, to the bodies involved, that non-branded foods embedded in media are not forgotten, as the regulation of the embedded foods have focused more on branded foods [74]. Additionally, we suggest to the relevant bodies focusing on the regulation of specifically, low nutritional value foods, as the overconsumption of these foods lead children to gain excessive weight. Lastly, execution variables (particularly modality) should also be considered by the corresponding bodies because they might be associated with the effects of foods embedded in entertainment media.

We believe that not only public bodies should take these results into account, but also media designers (such as cartoons designers), as they are the first implicated in the creation of the foods that appear as embedded in children’s entertainment media.

## 5. Conclusions

Considering the present systematic review and meta-analyses, foods embedded in children’s entertainment media are communication tools that influence the eating behaviors of children (both food choice and food intake). As the included studies focused on low nutritional value foods, the contribution to the main effects was mostly due to these types of foods. Unfortunately, children are frequently exposed to low nutritional foods embedded in media [14,15,16]. On the whole, the present study demonstrates the need for greater legislation regarding the foods embedded in entertainment media (cartoons, movies, TV shows, sitcoms, videogames, advergames, etc.) by policymakers, especially considering that those foods are low in nutritional value, as this is one of the main variables that will largely affect the problem of childhood obesity. Addressing the problem of childhood obesity requires a holistic approach that combines not only the present focus of this study, but also school-based interventions, parental practices, and other environmental factors [6], which are aimed at reducing the current obesogenic environment.

## Figures and Tables

**Figure 1 nutrients-12-00964-f001:**
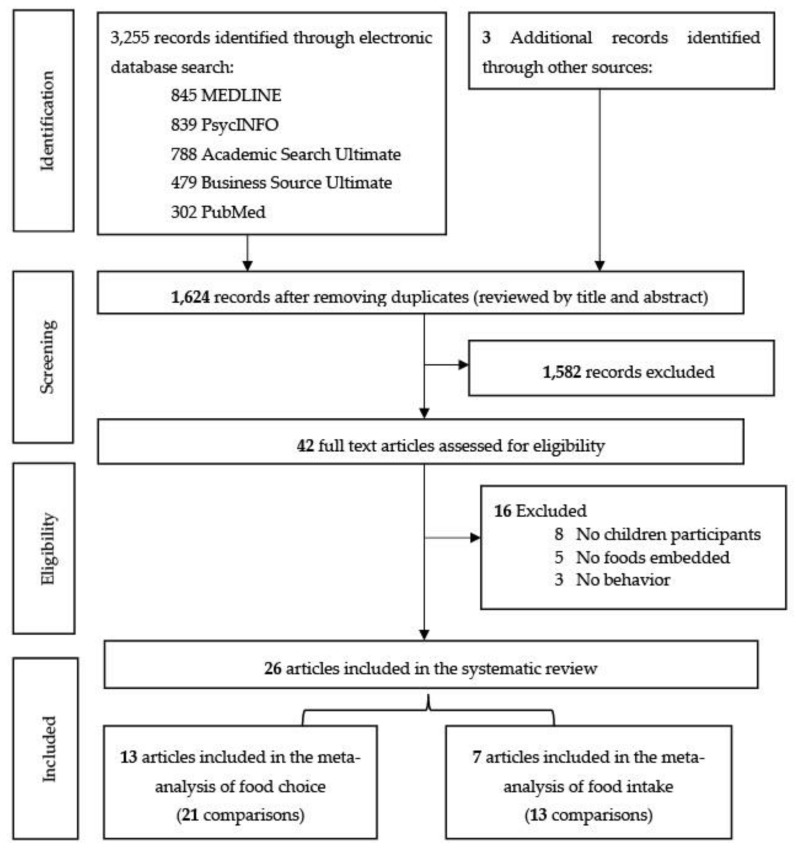
PRISMA diagram of search strategy.

**Figure 2 nutrients-12-00964-f002:**
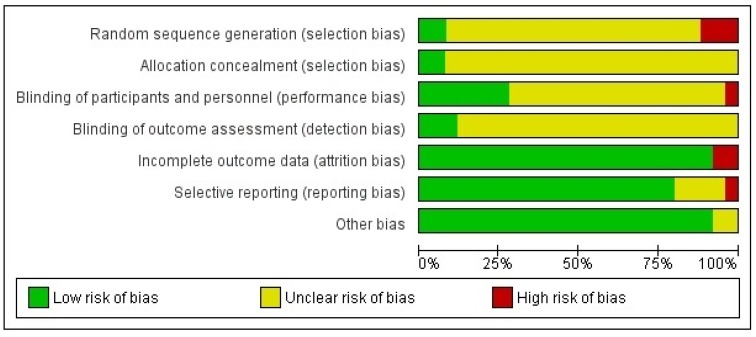
Risk of bias graph.

**Figure 3 nutrients-12-00964-f003:**
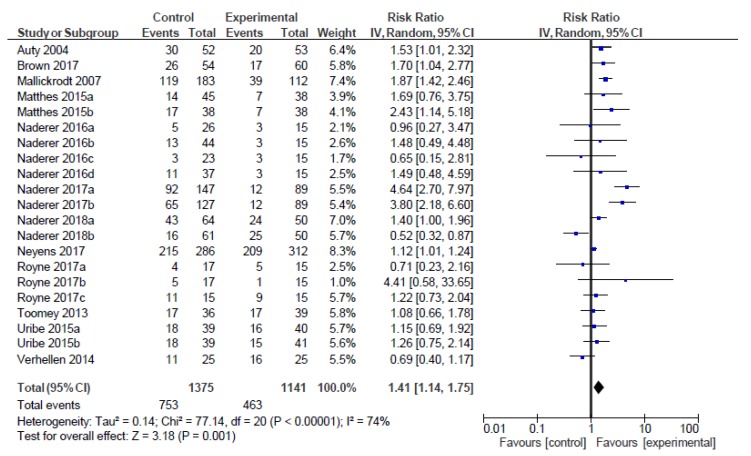
Forest plot of effect sizes on food choice.

**Figure 4 nutrients-12-00964-f004:**
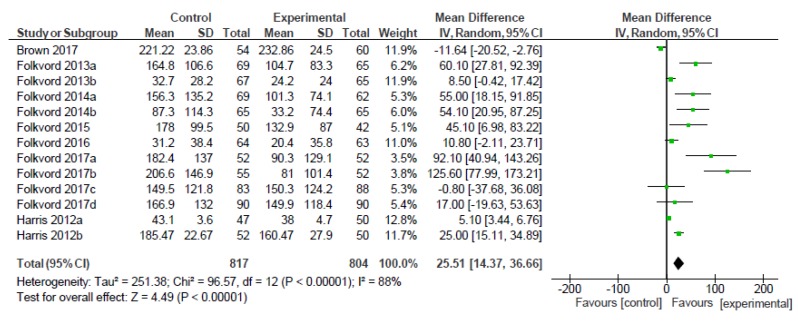
Forest plot of effect sizes on food intake.

**Figure 5 nutrients-12-00964-f005:**
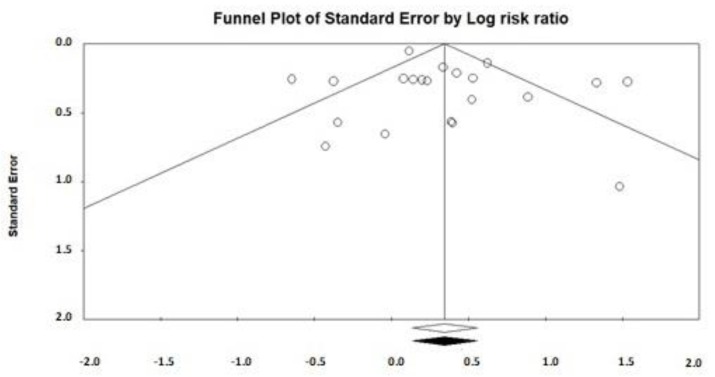
Funnel plot based on food choice.

**Figure 6 nutrients-12-00964-f006:**
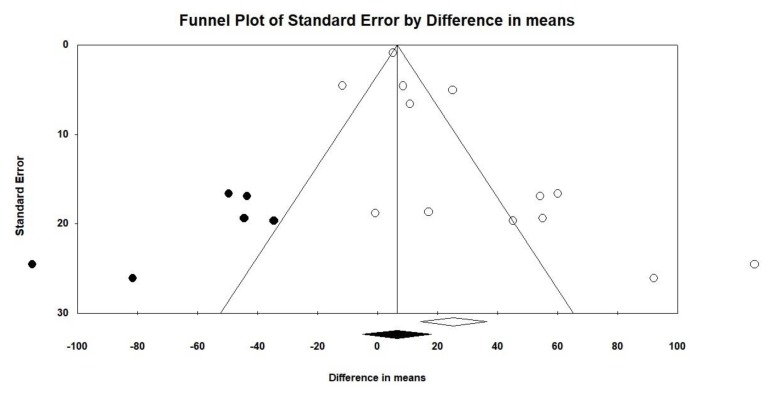
Funnel plot based on food intake.

**Table 1 nutrients-12-00964-t001:** Synthesis of characteristics of the included studies.

Characteristics	Included in The Systematic Review (26 Studies)	Included in The Meta-Analysis for Food Choice (13 Studies, 21 Comparisons)	Included in The Meta-Analysis for Food Intake (7 Studies, 13 Comparisons)
**Number of children** **Median mean age ^1^ (years)**	5747 9.2	2516 9	1621 8.9
**Median sample size**	139	79	127
**Study’s region performance**			
Europe	17	13	10
North America	6	5	3
South America	1	2	0
Asia	1	0	0
Australia	1	1	0
**Quality assessment**			
Low risk of bias	7	9	6
Middle risk of bias	16	10	6
High risk of bias	3	2	1
**Type of foods**			
Exclusively low nutritional value foods	14	18	10
Mixed (low and high nutritional value foods)	10	2	0
Exclusively high nutritional value foods	2	1	3
**Branding**			
Branded	17	16	12
Non-branded	8	5	1
Combined (branded and non-branded)	1	0	0
**Plot connection**			
Connected	13	15	0
Disconnected	7	6	13
No categorized	6	0	0
**Modality**			
Unimodal	8	8	0
Bimodal	10	11	1
No categorized	8	2	12
**Entertainment media used to embed the foods**			
Media that requires only watching			
Movies	5	8	1
TV shows	2	0	0
Cartoons (designed for the experiment)	2	6	0
Cartoons (real)	2	3	0
Sitcoms	0	1	0
Media that combines watching and playing			
Videogames (designed for the experiment)	4	0	0
Videogames (real)	0	0	0
Advergames (designed for the experiment)	5	0	10
Advergames (real)	5	3	2

^1^ Median was calculated once the mean age of the included studies was extracted.

**Table 2 nutrients-12-00964-t002:** Results of the moderator analyses of studies investigating the effects of foods embedded in children’s entertainment media on choice.

	Trials	RR	95% LL	95% UL	Exp. (N)	Cont. (N)	*p*-Value	*p*-Value for Interaction	*I* ^2^
**Type of foods**									
Low nutritional value foods	18	1.52	1.23	1.87	1280	1061	0.000	0.954	72.88
Mixed nutritional value foods	2	1.44	0.25	8.28	34	30	0.684		58.34
High nutritional value foods	1	−	−	−	−	−	−		−
**Branding**									
Branded	16	1.55	1.21	1.99	1,201	996	0.000	0.150	68.76
Non-branded	5	1.02	0.61	1.71	174	145	0.939		
**Plot connection**									
Disconnected	6	1.10	0.80	1.51	134	114	0.559	0.124	0.000
Connected	15	1.51	1.17	1.94	1241	1027	0.001		
**Modality**									
Unimodal	8	0.99	0.74	1.34	204	130	0.961	0.039	0.000
Bimodal	11	1.61	1.14	2.26	702	587	0.007		79.74
No categorized	2	−	−	−	−	−	−		
**Activity**									
Watching	18	1.48	1.13	1.95	881	692	0.005	0.734	68.82
Playing	3	1.19	0.77	1.83	494	449	0.444		87.19
**Age**									
Up to 5 years	−	−	−	−	−	−	−		
From 6 to 7 years	5	1.73	1.34	2.22	313	172	0.000	0.266	0.000
From 8 to 12 years	16	1.41	1.11	1.81	1062	969	0.005		77.79
**Quality assessment**									
Low risk of bias	9	1.26	0.89	1.78	390	289	0.197	0.692	53.86
Middle risk of bias	10	1.55	1.10	2.19	895	753	0.012		84.48
High risk of bias	2	1.36	0.88	2.11	90	99	0.172		37.67

Abbreviations: Risk Ratio (RR), Lower Limit (LL), Upper Limit (UL), Experimental (Exp.), Control (Cont.).

**Table 3 nutrients-12-00964-t003:** Results of the moderator analyses of studies investigating the effects of foods embedded in children’s entertainment media on intake.

	Trials	MD	95% LL	95% UL	Exp. (N)	Cont. (N)	*p*-Value	*p*-Value for Interaction	*I* ^2^
**Age**									
Up to 5 years	−	−	−	−	−	−	−		−
From 6 to 7 years	2	54.49	29.53	79.45	134	127	0.000	0.016	0
From 8 to 12 years	10	20.89	9.64	32.14	683	677	0.000		87.64
**Quality assessment**									
Low risk of bias	6	46.55	11.52	81.58	416	412	0.009	0.259	87.17
Middle risk of bias	6	24.70	10.14	39.25	347	332	0.001		85.46
High risk of bias	1	−	−	−	−	−	−	−	−

Abbreviations: Mean Difference (MD), Lower Limit (LL), Upper Limit (UL), Experimental (Exp.), Control (Cont.).

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
