# Peer review of "The Effects of Foods Embedded in Entertainment Media on Children’s Food Choices and Food Intake: A Systematic Review and Meta-Analyses"

_nutrients, 2020, doi:10.3390/nu12040964_

Round 1

Reviewer 1 Report

Thank you for giving me the opportunity to review this paper which is investigating how current experiments are exploring the effects of food placements in entertainment media. The paper presents an interesting and important topic. However, for the current version of the manuscript it is not really clear how the authors are contributing to the broad research field in this area. Comments are made below for each section of the manuscript:

Abstract

  • The authors describe that they conducted a systematic review as well as two meta analyses. However, they are “only” describing the results of the two meta analyses. To conduct a meta analysis it is of course necessary to conduct a systematic review.
  • Moreover, the abstract should be framed more precise. Now the abstract is quite lengthy and is still not describing the main goal of the study and the reason why these meta analyses are necessary.
  • Furthermore, the authors do not make clear what they define as entertaining media and what specific media formats the meta analyses include (e.g., advergames, TV shows, movie, social media, etc.). This should be already made clear in the abstract.
  • The authors state that overall “children exposed to food placements in entertainment media showed a higher risk of choosing the foods”. Since the authors are referring to unhealthy as well as to healthy food placements this formulation is confusing. Why should it be risky to choose a healthier food option after being exposed to healthy food placements? I would recommend to rephrase this sentence.

Introduction

  • In the first paragraph (line 40-48) the authors are describing videogames as new media formats. However, I would not say that videogames are new formats since they exist already since many, many years. Moreover, the authors state that food messages are also embedded “within other type of entertainment media content” without describing what these types should be. Also, in television programs embedded food placements are possible and studies already showed that these presentations can have an influence on children’s dietary behavior. So, from the first paragraph it is not really clear on which formats the authors focus.
  • The authors have to make a distinction between videogames and advergames. These are completely different formats. In line 55 the authors are saying that in videogames many food messages are embedded. However, the authors are citing a study focusing on advergames. Advergames and videogames are not the same.
  • From line 59 to line 63 the authors are describing why their study is important (e.g., health risk). However, this paragraph is in between the description of previous studies conducted in this area of research. Therefore, there is not really a common threat in the introduction section.
  • I would recommend not using the term food messages, since the described studies are focusing on food placements. The authors are switching between the terms food messages and food placements. However, these terms are describing different concepts. It is important to make clear to want concept the authors are referring to.
  • In line 58, the authors describe that food placements might affect children’s eating patterns. However, there exist already experimental studies as well as meta analyses investigating this relationship. These studies have to be cited.
  • From line 64 to 71 the authors describe meta analyses already conducted in this area of research. The authors should describe in more detail what the former studies investigated and on which formats these studies focused. With the current description it is not really clear what the contribution of the conducted study would be since there exist already meta analyses in this area of research.
  • Starting from line 82 to 115, the authors are describing on what aspects their meta analyses are focusing on and why. However, they are describing the results of the studies very superficial and it is not clear on which formats the studies are focusing on.
  • Moreover, as already stated form the abstract, the introduction does not describe at all on which formats the authors focus on and why. So, after the introduction it is not clear which formats fall in the category “entertainment media” and why the authors choose these formats. Social media for example can be also described as entertainment media. Without clarifying on which formats these meta analyses focus on it is not possible to evaluate the contribution of this study properly.
  • Overall it is not clear why the authors are conducting the study.

Materials and Methods

  • Also, in the method section, the authors are not defining what “entertainment media” are. Therefore, it is not possible to evaluate the meta analyses properly.
  • The authors describe that they categorized the variable age in two groups: Age under or over 9 years. However, the authors do not give a reason for that. Based on the development of children, it would make much more sense to divide the children due to their cognitive abilities (John, 1999). Moreover, the meta analyses conducted in this area are all not dividing the participants into two age groups.

Results

  • All figures within the manuscript are not really readable as well as different fonts are used.
  • In Table 1 the authors describe finally the media which were analyzed within the studies. However, they describe advergames as videogames which are two completely different media. Advergames are normally played online and display brands in a specific manner. Most of the studies included in this meta analyses are focusing on advergames. However, there exist already meta analyses analyzing this specific format (van’t Riet & Folkvord, 2018; DeSmet et al., 2014). Moreover, there are already meta analyses including advergames as well as TV shows (Russell, Croker, & Viner, 2018).

Discussion & Conclusion

  • Overall your discussion is comprehensible and I broadly agree with your conclusion. However, the writing up to this point does not do enough to convince me that your findings have actually pointed there.

Overall, the authors present an interesting study. However, with its current presentation it is not really clear what this study is contributing to the already big research field in this area. There exit already meta analyses for unhealthy and healthy advergames (van’t Riet & Folkvord, 2018; DeSmet et al., 2014) as well as for a combination of different embedded food advertising (Russell, Croker, & Viner, 2018) with children.

Furthermore, the paper needs substantial English proofing and revisions. The language is cumbersome and not easy to follow (e.g., line 54, line 64).

Literature:

Folkvord, F., & van‘t Riet, J. (2018). The persuasive effect of advergames promoting unhealthy foods among children: A meta-analysis. Appetite, 129, 245-251.

DeSmet, A., Van Ryckeghem, D., Compernolle, S., Baranowski, T., Thompson, D., Crombez, G., ... & Vandebosch, H. (2014). A meta-analysis of serious digital games for healthy lifestyle promotion. Preventive medicine, 69, 95-107.

Russell, S. J., Croker, H., & Viner, R. M. (2019). The effect of screen advertising on children's dietary intake: A systematic review and meta‐analysis. Obesity reviews, 20(4), 554-568.

Reviewer 2 Report

This was a very interesting paper that will make a nice contribution to the literature. That said, I have several suggestions for improvement:

  • In the Intro (and maybe also in the Abstract), I suggest the authors define more explicitly what they mean by "placement" - what counts as a placement and what doesn't. In the gaming world, advergames have been the focus of more studies than brand placements subtly included in non-food games. I'm assuming that the latter types of games were included in the sample but not the former - although I'm very unclear.

  • There has been other work on the impact of branding. See:

    van Reijmersdal, E. A., Rozendaal, E., & Buijzen, M. (2012). Effects of prominence, involvement, and persuasion knowledge on children's cognitive and affective responses to advergames. Journal of Interactive Marketing, 26, 33-42. doi:10.1016/j.intmar.2011.04.005
  • I'm confused as to why the authors only included studies published in English - especially when they're clearly based in Spain and presumably speak Spanish. Can further rationale be provided?

  • Why did you use age 15 as your inclusion threshold? Seems like an arbitrary choice.
  • If this were a journal with slower turnaround times or if the study hadn't been pre-registered, I would have asked for the following additions. In the event that the article is rejected, I recommend the authors consider them:
    • Ages 5, 7, and 12 are major cutoff points in terms of children's knowledge around advertising. The literature suggests that at each of these ages (approximately), children acquire more knowledge about persuasive intent, selling intent, and the like. As such, I would have liked to have seen the studies categorized into three age buckets, or for one of these ages to have been used as the cutoff, rather than the sample median.

    • It's really, really a shame that the authors didn't compare the effectiveness of passive media (e.g., TV) and active media (e.g., video games). This is a hot topic in the literature, with mixed evidence suggesting that active media are more impactful. Would have been a huge contribution to the literature to have tested this.

  • In cases where there were multiple comparisons within studies, was that non-independence accounted for?

  • Why were only 19 of the 25 identified articles included in the review?

  • If you have time, it would be nice to convert the text in 3.1 into a Table. That section was overly verbose.
  • I understand that the authors were using PRISMA, but I think it seems unfair to penalize / criticize these studies for essentially not double-blinding. That's highly unusual in media research in any topic area. Since the quality results from the meta-analytic models weren't that interesting or significant, I wonder if the authors could decrease attention to study quality?

  • While I appreciate the authors' integrity in pointing out when sample sizes are small and variance high, I suggest toning down the disqualifier language. It's undercutting your results too much.

  • Please add language to help readers interpret the results of your publication bias analyses, and add a key to the funnel plot differentiating between the white and black dots.

  • Dual-coding theory might be more appropriate to invoke than social cognitive theory.

  • In the Discussion, can you add more explanation around the plot connection, branding, and age findings? For age, why would age be significant for one outcome but not the other?

Minor editorial notes:

  • The term "moderator analyses" is more commonly used than "subgroup analyses".
  • Line 69: Change "related with" to "related to".
  • Revise lines 75-9 for clarity.
  • Add a citation to the end of line 96.
  • I don't understand the use of citation 34 on line 164.
  • Consider moving the sentence beginning with "In both cases, the comparison group..." to the section on study inclusion criteria.
  • Line 185: Add the word "not" between "kcal might" and "be a sensible".
  • Revise the sentence beginning on line 191 for clarity.
  • Consider another transition besides "Thus" on line 207. That sentence doesn't logically follow the preceding sentence; it's introducing a new type of reliability.
  • Line 256: I'm not sure what is meant by "qualitative".
  • Avoid "Besides" as a transition in English-language writing. English speakers and writers don't use "Besides" in a way that perfectly aligns with the way "Ademas" is used.
  • Find a way to clarify the sample sizes in your chart. The current column labels are confusing with unintuitive abbreviations.
  • Can you rephrase lines 331-333 to read less like Results section language?
  • Line 342: Delete "format" and make "placements" plural (i.e., add an 's').
  • Line 355: Replace "Nonetheless" with another transition word.
  • Line 384: Delete "an".

Round 2

Reviewer 2 Report

I applaud the authors on such a thorough, thoughtful, and speedy revision! Very impressive. Although I do have a few lingering questions/comments, at this point they're all very minor:

  • My intent in recommending the van Reijmersdal article was to buttress your argumentation around the impact of branding in games.

  • I think you can cut the sentence beginning with "However" on page 264. That's the only point you're dedicating so much space to a non-significant finding, and doing so seems unnecessary when the parallel finding ends up being significant in the next set of models.

  • Why don't you report when an individual category was significant, even when the interaction was not, for the intake moderators like you do for the choice moderators?

  • I still think you need to do more to interpret the publication bias findings for the readers. It seems like your analyses are implying that the intake effect would have nearly disappeared if you had located the "missing" studies. It's important for readers to understand that.
    • On a related note, I recommend expanding your discussion of why the choice findings might be more robust than the intake findings.

  • I'm not sure the reference to SCT on line 365 is correct.

  • Your interpretation of the study quality findings are a little misleading. You did find an effect for medium quality studies. Can you interpret that?

Editorial notes:

  • It seems like the authors may have used the "replace all" command to change "placements" to "food embedded". I suggest doing a careful read through to make sure that that is always grammatically and stylistically correct, and that the verb choice in the rest of the sentence corresponds to that phrasing choice.
  • On line 48, there should be a semi-colon or period between "task" and "there".
  • Replace "englobed" with something like "termed" or "labeled". 
  • The authors neglected to finish the sentence that currently ends on line 63.
  • I don't understand what the new sentence on line 153 is intending to convey.
  • Add "on" between "analysis" and "type" on line 181.
  • Change "was approximate to 12 years" to "was approximately 12".
  • Remove "the characteristic" on line 246.
  • Change "what" to "which"on line 299.
  • Delete the "to" in between "Regarding" and "the meta-analyses" on line 332.
  • Change the beginning of the sentence on line 355 to "However, when individual moderators are considered..."
  • Replace "appealing" with some other word on line 357.
  • The word "spend" should be plural on line 359.
  • I don't understand what the following means: "in defining foods embedded in children´s entertainment media legislation according to the type of food according to its healthiness."
  • Change "no" to "not" on slide 452.
